# Detection of microplastics in the feline placenta and fetus

Ilaria Ferraboschi[1], Francesca Canzolino[1], Elena Ferrari[1,2], Cristina Sissa[1], Matteo Masino[1]*, Matteo Rizzi[3], Simona Bussolati[3], Giuseppina Basini[3], Simone Bertini[3], Stefano Grolli[3], Roberto Ramoni[3], Francesco Di Ianni[3], Alessandro Vetere[3]*, Enrico Bigliardi[3]

1 Department of Chemistry, Life Science and Environmental Sustainability, University of Parma, Parma, Italy, 2 Istituto dei Materiali per l'Elettronica ed il Magnetismo, Consiglio Nazionale delle Ricerche (IMEM–CNR), Parma, Italy, 3 Department of Veterinary Science, University of Parma, Parma, Italy

* matteo.masino@unipr.it (MM); alessandro.vetere@unipr.it (AV)

## Abstract

The present study aimed to detect microplastics in feline placentas and fetuses in the early stage of pregnancy. For this study, 8 pregnant queens were evaluated. A standardized protocol for the digestion of biological matter was used, as well as a plastic-free approach for sample collection and manipulation. Microplastics were investigated by means of Raman spectroscopy, with the aim of identifying their composition. Four of eight animals were contaminated, with a total of 19 microplastics detected in both fetal and placental samples. Specifically, fetuses from cats 4 and 7 were contaminated, as were the placentas from cats 5, 6, and 7. This work demonstrates that microplastics can accumulate in feline placentas even at the early stage of pregnancy. Moreover, preliminary results of the presence of microplastics in feline fetuses are shown, suggesting that microplastics can cross the placental barrier.

## Introduction

During the XX century, plastic production reached more than 320 million tons per year, and more than 40% of plastic waste is used in single-use packaging; hence, plastic waste ends up littering the environment [1,2]. Atmospheric agents such as abrasion, waves and ultraviolet radiation in combination with bacteria degrade plastic fragments in micro- and nanosized particles [3,4]. Microplastics (MPs) are produced through the degradation of larger plastic materials into smaller fragments. These particles are pervasive across various ecosystems, including marine, freshwater, terrestrial, and atmospheric environments. MPs are detected in significant concentrations in oceans, rivers, groundwater, sediments, soils, sewage, and even in the air we inhale. Plastics exhibit high resistance to biodegradation due to their chemical stability; however, they can be broken down into micro- and nanoscale particles via mechanical abrasion and photochemical processes [5]. Nanoplastics (NPs), defined as plastic particles within the size range of 1 nm to 1 μm, share similar physicochemical properties and biological interactions with MPs. However, NPs have enhanced biological mobility and bioavailability, primarily due to their smaller size, allowing them to traverse biological membranes more effectively [5–8]. For the purposes of this paper, the term micro-nano-plastics (MNPs) will

**Data availability statement:** All relevant data are within the paper and its Supporting information files.

**Funding:** This research was supported by the Program "FIL-Quota Incentivante" of University of Parma and co-sponsored by Fondazione Cariparma. The work has benefited from the equipment and framework of the COMP-HUB Initiative, funded by the "Departments of Excellence program of the Italian Ministry for Education, University and Research (MIUR, 2018–2022). IF benefited of a PhD fellowship financed by PON R&I 2014-2020 (FSE REACT EU fundings). The funders had no role in study design, data collection and analysis, decision to publish, or preparation of the manuscript.

**Competing interests:** The authors have declared that no competing interests exist.

be employed to refer to plastic particles < 5 mm in size, encompassing both MPs and NPs. There has been global concern about microplastics in recent years because of their widespread distribution throughout seabeds and terrestrial [9] and airborne environments [10]. Moreover, microplastics have been found in food, and particularly in seafood [11,12] rather than in drinking water [13]. Microplastics have also been detected in the gastrointestinal tracts of marine animals [14] and in human stool [15], suggesting a bioaccumulation mechanism. Tissue accumulation of microplastics may lead to various adverse effects, such as immune responses [16], physical injury [17], inhibition of growth and development [18], metabolic disorders [19] and genotoxicity [20]. Another great concern arises from plastic intrinsic additives that are active at low concentrations, such as phthalates, bisphenol A and flame retardants [21]. The human and animal body are exposed to MNPs through three primary pathways: inhalation, ingestion, and dermal contact. Annual exposure to MPs alone is estimated to range between 74,000 and 121,000 particles per individual [22], with ingestion and inhalation identified as the main routes of entry. Since this estimate does not account for NPs, the actual exposure to MNPs is likely to be significantly higher [23,24]. Notably, MNPs have the ability to cross physiological barriers such as those in the lungs, gastrointestinal tract, and skin. Exposure to MNPs has demonstrated significant biological effects in rodent models, including a reduction in both sperm quantity and quality, as well as decreased testicular androgen synthesis. Additionally, circulating levels of testosterone and luteinizing hormone (LH) are diminished following MNP exposure. These findings suggest that MNPs may interfere with the pituitary-gonadotropin endocrine pathways, potentially impairing testicular function and sperm quality in male mammals [25]. Polystyrene micro-nano-plastics (PS-MNPs) are among the most studied in terms of female reproductive toxicity. In rodent models, oral exposure leads to PS-MNP accumulation in the ovaries and uterus, altering follicle dynamics, reducing ovarian weight, and affecting hormone levels [26,27]. These disruptions result in reduced fertility, with changes in estrous cycles, ovarian reserve, and litter sizes. The underlying mechanism is largely attributed to oxidative stress, evidenced by increased markers like malondialdehyde and higher rates of apoptosis in ovarian tissues. Growing evidence indicates that in utero exposure to microplastics negatively impacts fetal body weight and organ development [28]. Studies in mice show that fetuses exposed to MNPs exhibit significant skeletal muscle dysplasia and altered gene expression related to muscle development, lipid metabolism, and skin formation. These findings highlight the potential developmental risks of prenatal microplastic exposure [28]. A study lead by Li et al. [26] confirmed the presence of MPs in both domestic and fetal pig lung tissues from a natural environment, with domestic pig lungs showing higher MP concentrations (12 particles/g vs. 6 particles/g in fetal lungs). MPs were mostly fibrous, with polyamide dominating in domestic pigs and polycarbonate in fetal pigs. These findings suggest that MPs can be inhaled and potentially reach fetal tissues even without direct maternal exposure, unlike previous studies in rats and mice where MPs were detected in fetuses only after maternal exposure.

Recently, increasing attention has been given to the human uptake of MNPs, which are of particular interest for human health, but the related data remain limited [29]. In the last years, many studies demonstrated that microplastics can be found in the human placenta [5–7,30–33]. The placenta finely regulates the equilibrium between fetal and maternal compartments, acting as a crucial interface via distinct complex mechanisms [34]. The presence of microplastics in this organ and in this stage of life may lead to improper differentiation between nonself and self-compartments [35].

Moreover, the presence of microplastic was demonstrated also in human amniotic liquid [36] and carbon black particles were found in maternal blood, cordon blood and fetal organs [37], proving that particles can cross the placenta and reach the fetus.

To give a better insight on this topic, the present study aimed to assess firstly whether microplastics are present in the placenta and fetuses at an early stage of pregnancy. Cats were used as a model of a real in vivo exposure. A standardized protocol for the digestion of biological matter was used, as well as a plastic-free approach for sample collection and manipulation. Microplastics were investigated by means of Raman spectroscopy, with the aim of identifying their composition.

This work demonstrates that microplastics can accumulate in feline placentas even at the early stage of pregnancy. Moreover, preliminary results of the presence of microplastics in feline fetuses are shown, suggesting that microplastics can cross the placental barrier.

## Material and methods

### Animals

For this study, 8 pregnant queens were brought to the Obstetrics Unit at the Veterinary Teaching Hospital (OVUD) at the Department of Veterinary Medicine of the University of Parma. The total number of placentas and fetuses was 8. Foetal ages for the 8 pregnancies have been reported as follows: 15, 20, 27, 27, 28, 29, 30, and 30 days, with a mean age of 26.0 days. All queens were stray cats who came to the hospital for the population control program of the Emilia–Romagna Region. All cats were clinically healthy on physical examination, and the gestational ages were estimated to be between 15 and 30 days. This study was conducted in strict accordance with the recommendations outlined in the *Guide for the Care and Use of Laboratory Animals* of the National Institutes of Health. The experimental protocol was approved by the Ethical Committee for Animal Experimentation (ECAE) of the University of Parma, Italy (Protocol number 3/CESA/2022). All fetuses were humanely euthanized by intracardiac injection of embutramide, mebezonium iodide, and tetracaine (Tanax®, MSD Italia S.r.l,).

### Surgical procedure

All animals were premedicated with a combination of ketamine (5 mg/kg; Lobotor®, Acme S.r.l.), dexmedetomidine (7 mcg/kg; Dexdomitor®, Vetoquinol Italia S.r.l.) and butorphanol (0.2 mg/kg; Nargesic®, Acme S.r.l.) administered intramuscularly, followed by intravenous propofol (3 mg/kg; Proposure®, Boehringer Ingelheim Animal Health Italia S.p.A.) administration. A prophylactic injection of a combination of benzylpenicillin + streptomycin (2 ml/10 kg b.w.; Neotardocillina®, Vetoquinol Italia S.r.l.) was administered intramuscularly, and a single dose of robenacoxib (2 mg/kg; Onsior®, Elanco Italia S.p.A.) was administered subcutaneously. Lidocaine spray was applied to the arytenoid cartilage, and the cats were then intubated; anaesthesia was maintained with isoflurane (1.5%-2%) in oxygen. Ovariohysterectomy was performed conventionally through a ventral midline laparotomy. As soon as the ovaries and the gravid uterus were removed, the embryonic chambers were sectioned to divide the placentas and fetuses. Fetuses were humanely euthanized by intracoelomic injection of embutramide, mebezonium iodide, and tetracaine (Tanax®, Strada di Olgia Vecchia, Cd Milano Due, Palazzo Canova, 20054 Segrate (MI), Italy). All these components were immediately placed in single glass containers and frozen at -40 °C.

### Sample collection

In this study, only queens in the first half of pregnancy (within Day 30) were considered. The gestational age was determined by measuring the transverse diameter of the embryo vesicles [38,39] and performing ultrasound examination of the vesicles [38]. Ultrasound examination

was performed using a MyLabTM30Gold scanner (Esaote, Florence, Italy). The dimensions of the embryo vesicles, as well as the growth of the fetuses, differed according to gestational age.

According to previous studies [38,39], vesicles of 8-12 mm in diameter corresponded to a gestational age of approximately 15 days, vesicles of approximately 19-22 mm corresponded to a gestational age of 20 days, vesicles of approximately 27-30 mm corresponded to a gestational age of 25 days, and vesicles of 34-37 mm corresponded to a gestational age of 30 days.

Once ovariohysterectomy was performed, all the embryo vesicles were carefully dissected to collect placentas and fetuses separately. As the separation of the placenta from the endometrium during early pregnancy is particularly challenging due to the fragility of the tissues and the uneven number and distribution of placentas, we opted to standardize the sampling by including only one fetus per pregnancy. This approach was based on selecting cases where a complete and intact separation of the placenta was achieved, thereby minimizing variability and enhancing the reliability of the subsequent analyses. A plastic-free approach was adopted.

## Sample digestion

The digestion of placental and fetal samples was performed according to previously published protocols, with some modifications [21]. Briefly, both types of samples were incubated in a glass container with a 10% (w/v) KOH water solution. The ratio between the grams of sample and the volume of KOH was 1:8 (W/V). The water used for the preparation of the KOH solution had been previously filtered three times through a 1.6 μm pore-size filter membrane (Whatman GF/A) under vacuum. The samples were then stored at room temperature for 3 days. To avoid contamination of the samples, all the protective devices (laboratory coats, gloves, glasses, masks, etc.) employed were rigorously plastic free. Filtered water (1.6 μm) was used to prepare the 70% ethanol (V/V) solutions employed for cleaning bench working surfaces and for rinsing (three times) all the glassware and instruments (scalpels, scissors, and tweezers) used for the preparation of the samples. After the end of the incubation, the digested samples were filtered under vacuum through the same type of 1.6 μm pore-size filter used for the preparation of the water. The filters were dried at room temperature and then stored in a glass Petri dish until further processing. Three samples of the same KOH solution used for the digestion of the biological samples were subjected to the same procedure, and the resulting filters represented the blanks used to assess the eventual environmental contamination by plastic particles [40].

## Microplastic identification by Raman spectroscopy

MNPs characterization was performed using confocal micro-Raman spectroscopy immediately after digestion to avoid any contamination. Micro-Raman spectroscopy is one of the most used techniques for detecting and identifying MNPs in a great number of samples, ranging from environmental [41,42] to biological [43,44].

Raman spectra were acquired with a Horiba LabRAM HR Evolution Raman microspectrometer equipped with a liquid nitrogen-cooled CCD and a 600/mm grating blazed at 750 nm. Daily calibration was performed with a silicon slice using the 520.7 cm$^{-1}$ band.

The digested samples were deposited on white Whatman GF/A filters with a 1.6 μm pore size and were first imaged with a ×10 objective to locate all the particles. Two distinct sections of the filter surface were selected for examination to analyse several filters in a reasonable amount of time. A 50× magnification objective was then used to focus a 785 nm laser diode, with a spot radius of 2mm, on the sample and collect higher-quality Raman spectra. The acquisition parameters were modified to ensure that the best signal-to-noise ratio always started from the minimal laser power (below 0.1 mW) to prevent thermal degradation of the

materials. Only coloured particles smaller than 10 μm were considered for the analysis since they present a greater optical contrast with respect to the filter (which is white) and inorganic salts (such as $KCO_3$ and $KHCO_3$ which are transparent), and due to their size, they are the most likely to overtake the placenta and reach the fetus. Moreover, a blank filter was prepared and analysed under the same operating conditions as the specimen filters to track potential contamination during sample preparation. Raman spectra were collected for each particle, and identification of the pigment and/or of the polymer matrix was performed via comparison with spectra reported in the literature (the SLOPP library and peer-reviewed publications) and with KnowItAll software. Spectra were subjected to baseline correction before the fitting procedure to ensure data comparison between the acquired Raman spectra and those reported in the literature.

## Results and discussion

Eight animals before the thirtieth day of pregnancy, numbered from 1 to 8 based on the chronological order of the analysis, were used in this study. For each animal, a placenta and a fetus were exported, digested, and filtered following the procedure described in the Materials and Methods section. Two portions (700x500 mm) of each filter, selected with a 10x objective lens, were then analysed with a 50x objective lens to better detect and characterize the particles present.

Four of eight animals showed contamination of coloured particles of size < 10mm, with a total of 19 MNPs detected in both fetal and placental samples. Specifically, fetuses from cats 4 and 7 were contaminated, as were the placentas from cats 5, 6, and 7. Table 1 summarizes the main results, reporting the particle number, the cat and the organ (F = fetus, P = placenta)

**Table 1. List of the Main Features of the Detected Color Particles.** First column: particle number; Second column: information regarding where they were found (cat number; F: fetus and P: placenta); Third column: dimension of the particle; Fourth column: observed colour; Fifth column: identified pigments (n.d. = not determined); Sixth column: polymer matrix (n.d. = not determined).

| Particle | Provenience | Size | Colour | Pigment | Polymer matrix |
|---|---|---|---|---|---|
| 1 | Cat 4 – F | ~ 2 μm | Red | Mars Red+Anatase | n.d. |
| 2 | Cat 4 – F | ~ 2 μm | Red | Mars Red | PE |
| 3 | Cat 4 – F | ~ 2 μm | Red | Mars Red | PE |
| 4 | Cat 5 – P | ~ 5 μm | Orange | Raw Sienna | n.d. |
| 5 | Cat 5 – P | ~ 5 μm | Orange | Raw Sienna | n.d. |
| 6 | Cat 5 – P | ~ 3 μm | Orange | Goethite | PE |
| 7 | Cat 5 – P | ~ 2 μm | Red | Mars Red+Anatase | n.d. |
| 8 | Cat 5 – P | ~ 10 μm | Orange | Haematite | PE |
| 9 | Cat 5 – P | ~ 5 μm | Orange | Goethite | PE |
| 10 | Cat 5 – P | ~ 2 μm | Red | Mars Red+Anatase | PE |
| 11 | Cat 5 – P | ~ 5 μm | Orange | n.d. | PE |
| 12 | Cat 6 – P | ~ 5 μm | Brownish | Mars Red | n.d. |
| 13 | Cat 6 – P | ~ 10 μm | Brownish | Mars Red | n.d. |
| 14 | Cat 6 – P | ~ 5 μm | Dark brownish | Burnt Umber | n.d. |
| 15 | Cat 7 – F | ~ 5 μm | Blue | Alcian Blue | n.d. |
| 16 | Cat 7 – F | ~ 10 μm | Blue | Alcian Blue | n.d. |
| 17 | Cat 7 – F | ~ 5 μm | Blue | Alcian Blue | n.d. |
| 18 | Cat 7 – P | ~ 10 μm | Dark brownish | Burnt Umber | n.d. |
| 19 | Cat 7 – P | ~ 5 μm | Dark brownish | Burnt Umber | n.d. |

where they were detected, their size, the observed colour, and, where possible, the identified pigment and the polymer matrix.

Particle 11 is the only particle for which, despite its orange colour, the pigment could not be identified. However, the Raman spectrum (spectral region 850 cm$^{-1}$ -1450 cm$^{-1}$) of this particle, that is shown in Fig 1 together with the microscope image, is very interesting: the Raman peaks are indeed attributable to the polyethylene (PE) polymeric matrix. Specifically, C–C stretching is responsible for 1061 and 1130 cm$^{-1}$ peaks, 1297 cm$^{-1}$ is the CH$_2$ twisting peak, and the peaks at 1420, 1440, and 1464 cm$^{-1}$ are associated to the CH$_2$ bending [45]. Similar Raman peaks are identified also in particles 2, 3, 6, 8, 9, 10.

Fig 2 shows the microscope images and Raman spectra of the other coloured particles found in the placentas and fetuses. Raman spectra are shown in the spectral region where bands of dyes and pigments are detected (for particles 2, 3, 6, 8, 9, and 10, the peaks attributed to PE are reported in S1 Figure). Seven different types of dyes and additives were recognized: anatase, Mars Red, raw sienna, goethite, haematite, burnt umber and Alcian blue.

According to their Raman spectra, the analysed microparticles could be divided into seven groups based on their additives:

1. **_Particles 1, 7, and 10_** (Fig 2A): The Raman spectra of these particles are composite spectra resulting from the superposition of different components. The Raman bands match the bands of anatase (145, 198, 401, 640 cm$^{-1}$) and Mars Red (224, 291, 407, 496, 611 cm$^{-1}$). Anatase and rutile are two different forms of titanium dioxide and are known for their use in plastics as inexpensive, efficient, chemically and biologically inert additives [46]. Mars pigments, which were synthetically developed in the eighteenth century, are part of the iron oxide group of additives. They cover a large range of colours ranging from yellow to red, purple and black [47].

2. **_Particles 2, 3, 12, and 13_** (Fig 2B): The Raman spectra of these particles share the main peaks at 226, 293, 409, 499, and 611 cm$^{-1}$ with one pigment among the iron oxides, Mars Red, as determined by KnowItAll software.

3. **_Particles 4 and 5_** (Fig 2C): The peaks at 302, 396, 559, and 708 cm$^{-1}$ are superimposed onto those of the pigment known as the Raw Sienna. This additive contains manganese oxides

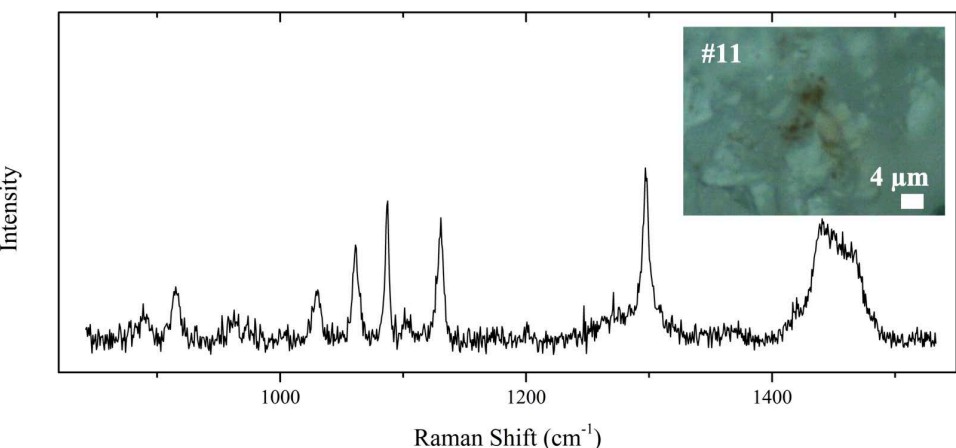

**Fig 1. Microscope Image and Raman Spectrum of Particle 11.** Peaks at 1061, 1130 (C–C Stretching), and 1297 (CH$_2$ Twisting) and the peaks at 1420, 1440, and 1464 cm$^{-1}$ (CH$_2$ Bending) are compatible with the presence of the PE Polymeric Matrix. The same peaks were also observed for particles 2, 3, 6, 8, 9, and 10.

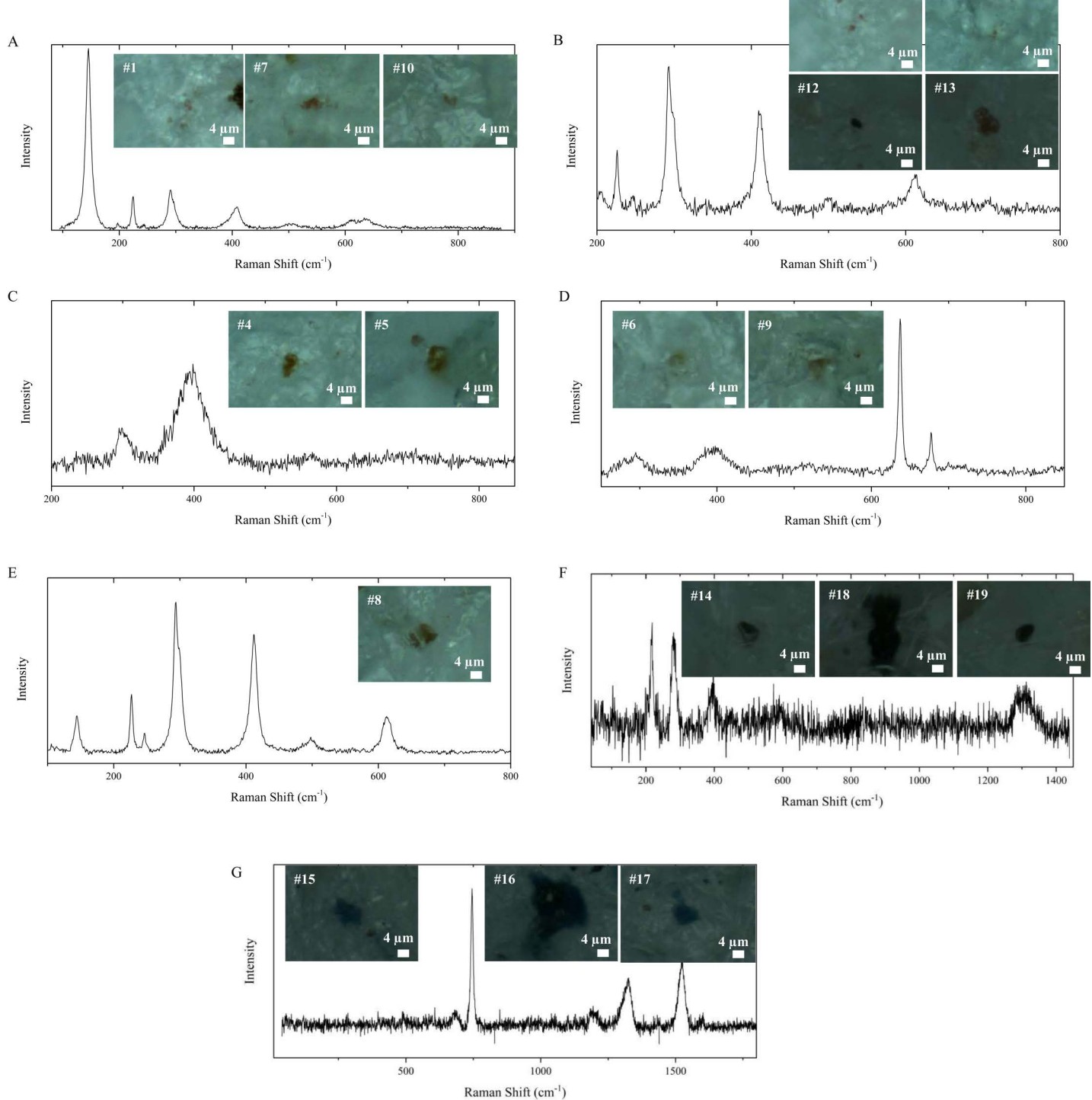

**Fig 2. Microscope Images and Raman Spectra of Colored Particles Smaller Than 10mm Detected in Feline Placentas and Fetuses.**

and is also rich in iron hydroxides, which are responsible for the brownish-yellow shades [47].

4. **Particles 6, 9** (Fig 2D): The Raman spectra of these particles are composite spectra resulting in the sum of the polymer matrix and the pigment bands. In particular, the peaks at 290 and 392 cm⁻¹ are assigned to the pigment iron hydroxide oxide yellow (goethite) [48].

5. **Particle 8** (Fig 2E): The peaks at 226, 247, 294, 412, 497, and 612 cm⁻¹ are attributed to a pigment classified as an iron oxide, haematite ($\alpha$-$Fe_2O_3$) [48].

6. **Particles 14, 18, and 19** (Fig 2F): Raman peaks at 216, 276, 387, and 590 cm⁻¹ are ascribed to a common iron oxide pigment called Burnt Umber. Generally, umber contains manganese oxide along with iron oxides in a fraction that makes it different from ochres and siennas [47].

7. **Particles 15, 16, and 17** (Fig 2G): The bands at 683, 745, 1198, 1328, 1432, 1525, and 1601 cm⁻¹ are related to Raman signals of a phthalocyanine blue pigment, Alcian blue, according to KnowItAll software.

Under the same experimental conditions, blank filters did not show the presence of coloured particles with a Raman spectrum compatible with the presence of plastic matrices or pigments.

Out of the 16 samples analysed (8 placentas and 8 fetuses), 5 were found contaminated by coloured particles. The 5 contaminated samples belong to 4 animals. 6 coloured particles were found in fetal samples while 13 were found in placentas. Through the comparison with spectra reported in the literature (the SLOPP library and peer-reviewed publications) and using the KnowItAll software, it was possible to identify the dye in 18 particles and the backbone of the polymeric matrix in 7 particles (out of 19). These 7 particles are classified as MNPs, while the remaining 12 particles are classified as suspect MNPs. Since Raman response is stronger for organic dyes then polymer matrices, it is common [30,44,49] to detect the pigment but not the polymer. For the same reason, it is not surprising that the polymer matrix was always identified as PE. PE is structurally composed by the repetition of the $CH_2$ motif, a building block present in the backbone of several polymers such as polypropylene, nylon 6, polyethylene terephthalate, etc. Since the backbone is the most repeated feature in a polymer, it is reasonable that the peaks associated with the $CH_2$ motif arise the strongest signals of the polymeric matrix, leading to the identification of those signal as PE.

It is worth noting that the methodology applied underestimates the real number of MNPs, possibly leading to false negatives. Only coloured particles were considered in our study since they present a better contrast compared to the filter and to the inorganic salts derived from the digestion process (KOH can react with atmospheric $CO_2$ generating $KHCO_3$ and $K_2CO_3$). Moreover, only particles between 1 and 10 mm were selected since they are more likely to be transported by the bloodstream. Nonetheless, the aim of this work was to determine whether MNPs can be found in the early stage of pregnancy and fetuses and not to quantify them. Therefore, even if underestimated, coloured particles were found in 2 fetuses out of 8, and in 2 particles found in fetuses, the polymeric matrix was identified by Raman spectroscopy, hinting that MNPs can overcome the placental barrier and reach the fetus even in the early stages of pregnancy.

The high production and consumption of plastics is a serious and practical issue. MNPs are pervasive across global ecosystems, are capable of contaminating water and animals and can be detected in the internal organs of the human body. MNPs contamination in food can be the result of airborne particles or equipment used in the processes of food production or packaging [50]. A recent study demonstrated that packaging could be a source of contamination in

products and that highly processed products contain more MNPs/g than minimally processed products; for example, chicken nuggets (highly processed) contain more MNPs than chicken breasts (minimally processed) [50]. Another investigation suggested that bottled drinking water contains higher levels of MNPs than tap water [51]. Packaging may be a cause of contamination of food and beverages. Today, it is becoming increasingly clear that MNPs are present throughout the entire human nutrition system. Recent studies have identified MNPs in human blood (>700 nm), lungs (>3 mm), heart (20-50 mm) and heart (1 to 469 μm) indicating the pervasive nature of microplastic contamination across various human tissues and organs [30,52–54]. In our study, we demonstrated for the first time that MNPs can accumulate in feline placentas at the early stage of pregnancy, and, even more interestingly, also in fetuses. The presence of MNPs in fetal tissue suggests possible consequences for fetal development that are currently unproven. The dimensions of all the MNPs detected were ≤ 10 mm, which is compatible with transportation via the bloodstream.

It has been reported that microplastics are present in the air within various indoor environments [55]. To address this concern, uteri were carefully washed with sterile saline solution prior to dissection to minimize the risk of external contamination. Furthermore, throughout all stages of processing, plastic instruments were never used, making environmental contamination of the samples highly unlikely. The placenta plays a vital role in the transport of nutrients and in the regulation of immunity, in which the regulation of the maternal immune response protects the fetus from the mother's immune aggression. Some complications of pregnancy, such as miscarriages, preeclampsia, and premature birth defects, can be attributed to alterations in immune balance. In a recent study [56], it was demonstrated that polystyrene microplastics (PS-MPs) can potentially cause adverse effects on pregnancy by negatively influencing the immune balance. Furthermore, following PS-MP exposure, the proportions of immune cells in the peripheral blood, spleen and placenta are altered [57]. Other studies have shown that in pregnant rats exposed to nanoplastics, the percentage of embryonic resorption increased significantly [57]. It is interesting to emphasize that, in our study, the animals lived free in nature and had a partly commercial and partly natural diet. This shows how the intake of MNPs can derive directly from the commercial diet and from the capture of prey. In a recent preclinical study, MNPs emerged as potential risk factors for cardiovascular disease. Also, this study underscores the pressing need to limit or eliminate the use of plastics and encourages scientists to explore and develop viable, sustainable alternatives. These findings should serve as a wake-up call for policymakers and industries to prioritize strategies aimed at mitigating plastic pollution to protect both environmental and biological systems. In conclusion, MNPs represent a risk to human and animal health and have now entered the food chain. Several studies have demonstrated the toxicity of MNPs to the reproductive system [55], and the identification of MNPs in fetuses raises the question of their potential toxicity to fetal development. Future research will be necessary to understand the fetal organs in which MNPs accumulate and the possible complications of pregnancy.

## Author contributions

**Conceptualization:** Ilaria Ferraboschi, Cristina Sissa, Giuseppina Basini, Simone Bertini, Stefano Grolli, Enrico Bigliardi.

**Data curation:** Ilaria Ferraboschi, Francesca Canzolino, Elena Ferrari, Simona Bussolati, Stefano Grolli, Francesco Di Ianni, Alessandro Vetere, Enrico Bigliardi.

**Investigation:** Francesca Canzolino, Elena Ferrari, Cristina Sissa, Matteo Masino, Matteo Rizzi, Simona Bussolati, Giuseppina Basini, Simone Bertini, Stefano Grolli, Roberto Ramoni, Francesco Di Ianni, Enrico Bigliardi.

**Methodology:** Ilaria Ferraboschi, Francesca Canzolino, Elena Ferrari, Cristina Sissa, Matteo Masino, Simona Bussolati, Giuseppina Basini, Simone Bertini, Stefano Grolli, Roberto Ramoni.

**Validation:** Francesca Canzolino.

**Visualization:** Stefano Grolli, Roberto Ramoni, Francesco Di Ianni.

**Writing – original draft:** Ilaria Ferraboschi, Cristina Sissa, Giuseppina Basini, Enrico Bigliardi.

**Writing – review & editing:** Ilaria Ferraboschi, Cristina Sissa, Alessandro Vetere.

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
