## [Decision Letter · Decision Letter 0]

2 Jan 2025

PONE-D-24-55176Detection of microplastics in the feline placenta and fetusPLOS ONE

Dear Dr. Vetere,

Thank you for submitting your manuscript to PLOS ONE. After careful consideration, we feel that it has merit but does not fully meet PLOS ONE’s publication criteria as it currently stands. Therefore, we invite you to submit a revised version of the manuscript that addresses the points raised during the review process.

We look forward to receiving your revised manuscript.

Kind regards,

Yousuf Dar Jaffer

Academic Editor

PLOS ONE

**Journal Requirements:**

Reviewers' comments:

Reviewer's Responses to Questions

**Comments to the Author**

1. Is the manuscript technically sound, and do the data support the conclusions?

Reviewer #1: Yes

Reviewer #2: Yes

2. Has the statistical analysis been performed appropriately and rigorously? 

Reviewer #1: N/A

Reviewer #2: Yes

3. Have the authors made all data underlying the findings in their manuscript fully available?

Reviewer #1: Yes

Reviewer #2: Yes

4. Is the manuscript presented in an intelligible fashion and written in standard English?

Reviewer #1: Yes

Reviewer #2: Yes

5. Review Comments to the Author

**Reviewer #1: ** The work by Ferraboschi et al. describes a very worrisome finding, i.e. the presence of microplastics in foetuses and placentae of stray cats. The object of the investigation is very innovative and interesting. Unwanted feline pregnancies, surgically interrupted through ovariohysterectomy respecting ethical issues, were used to detect microplastic contamination.

The work is rather clearly written, with some repetitions that should be deleted (Page 7, lines 115-117 and again lines 130-131) (Page 7, lines 128-129 and again on Page 8, lines 145-146).

My doubts:

The authors should clarify why a single foetus with his placenta was included from each pregnancy and what criterion was used to select it (always the same position within the uterus?).

Foetal age should be reported for the 8 pregnancies, notwithstanding the fact that no gestation age-related condition can be tested when only 8 pregnancies are included.

I wonder how an intracardiac injection can be done in 15-day embryos/foetuses (see Knospe (2002) Periods and Stages of the Prenatal Development of the Domestic Cat).

Is it true that a coloured particle for which the polymer matrix cannot be identified should be more correctly identified as a ‘suspect’ MNP. If so, this should be reported, modifying the results from line 261 to 271, Page 15.

I would add some comments on the possibility of contamination (see your Reference n 32 on human placentae collected at C-sections).

On page 15 (line 261) I do not understand where number 5 comes from: four pregnancies were contaminated by coloured MNPs (cats 4,5,6 and 7). In two cases (cat 4 foetus and cat 5 placenta) PE was detected. I cannot see 5.

On page 16 (lines 290-291), there is a wrong reference number and a wrong citation: if the correct number is 13 and not 31 (please, add brackets), tap water resulted contaminated with higher levels of MNPs than bottled water.

I would stress in the conclusions the alarming findings of this work, that should be a stimulus to

drastically limit/eliminate the use of plastic and, for scientists, to find valid alternatives to it.

Specific comments

Page 4, line 58: ‘(MPs’ delete ‘(‘

Page 7 line 121-122: Propofol is not part of premedication, but it is used for anaesthesia induction.

Page 12, line 212: ‘is’ instead of ‘in’

Line 294: do the two ranges refer to the heart? It is not clear.

Table 1: The columns do not correspond to what is indicated in the legend

Last: Why two corresponding authors?! It is rather unusual

**Reviewer #2:**  The manuscript was well written and this can be published in esteemed journal of PLOS ONE.------------------------------------------------------------------------------------------------------------------

6. PLOS authors have the option to publish the peer review history of their article (what does this mean? ). If published, this will include your full peer review and any attached files.

**Do you want your identity to be public for this peer review?** For information about this choice, including consent withdrawal, please see our Privacy Policy .

Reviewer #1: No

Reviewer #2: **Yes: ** Akbar Abbasi

---

## [Author Response · Author response to Decision Letter 1]

10 Feb 2025

Reviewer Questions and Responses

Question Response

The authors should clarify why a single foetus with his placenta was included from each pregnancy and what criterion was used to select it (always the same position within the uterus?). The separation of the placenta from the endometrium during an early stage of pregnancy is particularly challenging due to the fragility of the tissues, as well as the uneven number and distribution of placentas. To ensure uniformity in the samples, it was decided to use only one fetus for which complete separation of the placenta was successfully achieved. Line 150-156

Foetal age should be reported for the 8 pregnancies, notwithstanding the fact that no gestation age-related condition can be tested when only 8 pregnancies are included. Foetal ages for the 8 pregnancies have been reported as follows: 15, 20, 27, 27, 28, 29, 30, and 30 days, with a mean age of 26.0 days. (Line 110-111)

I wonder how an intracardiac injection can be done in 15-day embryos/foetuses (see Knospe (2002) Periods and Stages of the Prenatal Development of the Domestic Cat). Thank you for your important observation. The injections were performed in the thoracic region in the bigger fetuses; however, given the extremely small size of the fetuses and the inability to confirm the actual distribution of the drug within the circulatory system, it is more accurate to refer to these as intracoelomic injections. We have updated the phrasing accordingly.

Is it true that a coloured particle for which the polymer matrix cannot be identified should be more correctly identified as a ‘suspect’ MNP. If so, this should be reported, modifying the results from line 261 to 271, Page 15. We thank the referee for this important comment. We agree that the colored particles for which we were unable to identify the polymer matrix should be classified as suspect MNPs, and we have revised the discussion accordingly. Furthermore, we have moved the discussion about the Raman spectra of dyes/polymer matrix to this paragraph.

I would add some comments on the possibility of contamination (see your Reference n 32 on human placentae collected at C-sections). Thank you for your comment. Throughout all stages of processing, plastic instruments were never used. The laboratory environments were controlled. Digestion procedures were performed under a fume hood. It has been reported that microplastics are present in the air within various indoor environments (Zhang et al., 2020). To address this concern, uteri were carefully washed with sterile saline solution prior to dissection to minimize the risk of external contamination. We added few lines in the discussion section. Lines 315-320

On page 15 (line 261) I do not understand where number 5 comes from: four pregnancies were contaminated by coloured MNPs (cats 4,5,6 and 7). In two cases (cat 4 foetus and cat 5 placenta) PE was detected. I cannot see 5. The number of analysed sample is 16 (8 placentas and 8 fetuses). Out of these, we found 5 contaminated samples, belonging to 4 animals (in one case we found that both placenta and fetus were contaminated). In the text, we specified that the 5 samples belong to 4 animals.

On page 16 (lines 290-291), there is a wrong reference number and a wrong citation: if the correct number is 13 and not 31 (please, add brackets), tap water resulted contaminated with higher levels of MNPs than bottled water. Thank you for your comment. The correct reference was added.

I would stress in the conclusions the alarming findings of this work, that should be a stimulus to drastically limit/eliminate the use of plastic and, for scientists, to find valid alternatives to it. Thank you for your suggestion. Added at line 311-315.

Page 4, line 58: ‘(MPs’ delete ‘(‘ Done

Page 7 line 121-122: Propofol is not part of premedication, but it is used for anaesthesia induction. Thank you for your comment. Corrected. Line 142-143

Page 12, line 212: ‘is’ instead of ‘in’. Corrected

Line 294: do the two ranges refer to the heart? It is not clear. The typo was corrected

Table 1: The columns do not correspond to what is indicated in the legend. Corrected. In the legend, columns 2 and 3 were mistakenly swapped.

Why two corresponding authors?! It is rather unusual. Given that the work was carried out by researchers from different areas of expertise (clinical and spectroscopic laboratory), it was decided to designate two corresponding authors, each representing their respective field of expertise.

---

## [Editor Report · Decision Letter 1]

18 Feb 2025

PONE-D-24-55176R1Detection of microplastics in the feline placenta and fetusPLOS ONE

Dear Dr. Vetere,

Thank you for submitting your manuscript to PLOS ONE. After careful consideration, we feel that it has merit but does not fully meet PLOS ONE’s publication criteria as it currently stands. Therefore, we invite you to submit a revised version of the manuscript that addresses the points raised during the review process.

We look forward to receiving your revised manuscript.

Kind regards,

Yousuf Dar Jaffer

Academic Editor

PLOS ONE

Additional Editor Comments:

Dear Alessandro,

At this stage, I still see the manuscript has a scope of improvement. Please address the comments from reviewer 1 except the last comment on having two corrosponding authors. This decision relies on publisher.

Best regards,

Jaffer

---

## [Author Response · Author response to Decision Letter 2]

18 Feb 2025

Question: The authors should clarify why a single fetus with his placenta was included from each pregnancy and what criterion was used to select it (always the same position within the uterus?).

Answer: The separation of the placenta from the endometrium during an early stage of pregnancy is particularly challenging due to the fragility of the tissues, as well as the uneven number and distribution of placentas. To ensure uniformity in the samples, it was decided to use only one fetus for which complete separation of the placenta was successfully achieved (Lines 150-156).

Question: Foetal age should be reported for the 8 pregnancies, notwithstanding the fact that no gestation age-related condition can be tested when only 8 pregnancies are included.

Answer: Foetal ages for the 8 pregnancies have been reported as follows: 15, 20, 27, 27, 28, 29, 30, and 30 days, with a mean age of 26.0 days (Lines 110-111).

Question: I wonder how an intracardiac injection can be done in 15-day embryos/foetuses (see Knospe (2002) Periods and Stages of the Prenatal Development of the Domestic Cat).

Answer: Thank you for your important observation. The injections were performed in the thoracic region in the bigger fetuses; however, given the extremely small size of the fetuses and the inability to confirm the actual distribution of the drug within the circulatory system, it is more accurate to refer to these as intracoelomic injections. We have updated the phrasing accordingly.

Question: Is it true that a coloured particle for which the polymer matrix cannot be identified should be more correctly identified as a ‘suspect’ MNP. If so, this should be reported, modifying the results from line 261 to 271, Page 15.

Answer: We thank the referee for this important comment. We agree that the colored particles for which we were unable to identify the polymer matrix should be classified as suspect MNPs, and we have revised the discussion accordingly. Furthermore, we have moved the discussion about the Raman spectra of dyes/polymer matrix to this paragraph.

Question: I would add some comments on the possibility of contamination (see your Reference n 32 on human placentae collected at C-sections).

Answer: Thank you for your comment. Throughout all stages of processing, plastic instruments were never used. The laboratory environments were controlled. Digestion procedures were performed under a fume hood. It has been reported that microplastics are present in the air within various indoor environments (Zhang et al., 2020). To address this concern, uteri were carefully washed with sterile saline solution prior to dissection to minimize the risk of external contamination. We added a few lines in the discussion section (Lines 315-320).

Question: On page 15 (line 261) I do not understand where number 5 comes from: four pregnancies were contaminated by coloured MNPs (cats 4,5,6 and 7). In two cases (cat 4 foetus and cat 5 placenta) PE was detected. I cannot see 5.

Answer: The number of analyzed samples is 16 (8 placentas and 8 fetuses). Out of these, we found 5 contaminated samples, belonging to 4 animals (in one case we found that both placenta and fetus were contaminated). In the text, we specified that the 5 samples belong to 4 animals.

Question: On page 16 (lines 290-291), there is a wrong reference number and a wrong citation: if the correct number is 13 and not 31 (please, add brackets), tap water resulted contaminated with higher levels of MNPs than bottled water.

Answer: Thank you, corrected.

Question: I would stress in the conclusions the alarming findings of this work, that should be a stimulus to drastically limit/eliminate the use of plastic and, for scientists, to find valid alternatives to it.

Answer: Thank you for your suggestion. Added at line 311-315.

Question: Page 4, line 58: ‘(MPs’ delete ‘(‘

Answer: Done

Question: Page 7 line 121-122: Propofol is not part of premedication, but it is used for anaesthesia induction.

Answer: Thank you for your comment. Corrected. Line 142-143

Question: Page 12, line 212: ‘is’ instead of ‘in’.

Answer: Corrected

Question: Line 294: do the two ranges refer to the heart? It is not clear.

Answer: The typo was corrected

Question: Table 1: The columns do not correspond to what is indicated in the legend.

Answer: Corrected. In the legend, columns 2 and 3 were mistakenly swapped.

Question: Why two corresponding authors?! It is rather unusual.

Answer: Given that the work was carried out by researchers from different areas of expertise (clinical and spectroscopic laboratory), it was decided to designate two corresponding authors, each representing their respective field of expertise.

---

## [Editor Report · Decision Letter 2]

23 Feb 2025

Detection of microplastics in the feline placenta and fetus

PONE-D-24-55176R2

Dear Dr. Alessandro Vetere

We’re pleased to inform you that your manuscript has been judged scientifically suitable for publication and will be formally accepted for publication once it meets all outstanding technical requirements.

Kind regards,

Yousuf Dar Jaffer

Academic Editor

PLOS ONE
---

## [Editor Report · Acceptance letter]

PONE-D-24-55176R2

PLOS ONE

Dear Dr. Vetere,

I'm pleased to inform you that your manuscript has been deemed suitable for publication in PLOS ONE. Congratulations! Your manuscript is now being handed over to our production team.

Kind regards,

on behalf of

Dr. Yousuf Dar Jaffer

Academic Editor

PLOS ONE